# Clinical Efficacy of Fidaxomicin and Oral Metronidazole for Treating *Clostridioides difficile* Infection and the Associated Recurrence Rate: A Retrospective Cohort Study

**DOI:** 10.3390/antibiotics12081323

**Published:** 2023-08-16

**Authors:** Nobuaki Mori, Jun Hirai, Wataru Ohashi, Nobuhiro Asai, Yuichi Shibata, Hiroshige Mikamo

**Affiliations:** 1Department of Clinical Infectious Diseases, Aichi Medical University, 1-1 Yazakokarimata Nagakute-shi, Aichi 480-1195, Japan; 2Department of Infection Prevention and Control, Aichi Medical University, 1-1 Yazakokarimata Nagakute-shi, Aichi 480-1195, Japan; 3Division of Biostatistics, Clinical Research Center, Aichi Medical University, 1-1 Yazakokarimata Nagakute-shi, Aichi 480-1195, Japan

**Keywords:** *Clostridioides difficile* infection, efficacy, fidaxomicin, oral metronidazole, treatment failure

## Abstract

*Clostridioides difficile* infection (CDI) has significant implications for healthcare economics. Although clinical trials have compared fidaxomicin (FDX) and vancomycin, comparisons of FDX and oral metronidazole (MNZ) are limited. Therefore, we compared the therapeutic effects of FDX and oral MNZ. Patients diagnosed with CDI between January 2015 and March 2023 were enrolled. Those treated with oral MNZ or FDX were selected and retrospectively analyzed. The primary outcome was the global cure rate. Secondary outcomes included factors contributing to the CDI global cure rate; the rate of medication change owing to initial treatment failure; and incidence rates of clinical cure, recurrence, and all-cause mortality within 30 days. Of the 264 enrolled patients, 75 and 30 received initial oral MNZ and FDX treatments, respectively. The corresponding CDI global cure rates were 53.3% and 70% (*p* = 0.12). In multivariate analysis, FDX was not associated with the global cure rate. In the MNZ group, 18.7% of the patients had to change medications owing to initial treatment failure. The FDX group had a higher clinical cure rate and lower recurrence rate than the MNZ group, although not significant. However, caution is necessary owing to necessary treatment changes due to MNZ failure.

## 1. Introduction

*Clostridioides difficile* infection (CDI) stands as a major healthcare-associated ailment, inflicting substantial morbidity, mortality, and economic burden on a global scale [1]. In 2011, the US reported 453,000 initial cases of CDI, with 29,300 associated deaths [2]. Between 2011 and 2017, the incidence of CDI declined primarily because of reduced healthcare-associated infections [3]. Nevertheless, in 2019, the Centers for Disease Control and Prevention in the US designated *C. difficile* as one of the most serious public health-threatening pathogens, necessitating urgent and aggressive action. Additionally, the economic impact of CDI is noteworthy, with a recent study in the US estimating the costs of CDI and CDI recurrence during a 6-month follow-up period to be USD 24,205 (95% confidence interval (CI): USD 23,436–USD 25,013) and USD 10,580 (95% CI: USD 8849–USD 12,446), respectively [4].

According to the CDI guidelines published by the Infectious Diseases Society of America (IDSA) and the Society for Healthcare Epidemiology of America (SHEA) in 2021, fidaxomicin (FDX) is recommended as the first-line therapy for initial CDI [5]. These updated recommendations highlight the superior beneficial effects and safety of FDX compared with those of vancomycin (VCM) based on clinical studies [6,7]. The IDSA/SHEA guidelines do not offer guidance on the use of metronidazole (MNZ), which was previously advocated as first-line therapy for CDI. Nonetheless, MNZ has long been employed in CDI treatment due to its cost-effectiveness compared to VCM and its reduced likelihood of promoting VCM-resistant organisms. Japanese and Australian CDI guidelines recommend MNZ as the first-line therapy for non-severe CDI [8,9]. 

A retrospective nationwide cohort study demonstrated no improvement in treatment failure or probable recurrence between the pre- and post-guideline cohorts in the US, wherein MNZ usage was reduced [10]. Unfortunately, real-world comparative studies assessing the clinical efficacy of MNZ and FDX, along with their associated recurrence rates, remain limited. Thus, further research is necessary to determine the optimal treatment strategy for CDI.

This study aimed to comprehensively evaluate the clinical efficacy of FDX and oral MNZ in the treatment of CDI and the associated recurrence rates. By directly comparing these two treatment modalities, we aimed to elucidate the potential advantages and disadvantages of each regimen, which might aid clinicians in making informed decisions regarding CDI management.

## 2. Results

During the study period, 264 patients were assessed for eligibility (Figure 1). Of these, 166 were excluded based on the exclusion criteria. Thus, 105 patients were included in this study. Of these, 75 and 30 were assigned to the oral MNZ and FDX groups, respectively. The demographic and baseline characteristics of the study population are shown in Table 1. In both groups, the median age was 76 years, and most patients were male. The two groups were well matched in terms of baseline characteristics such as age, sex, and comorbidities. Patient conditions at the time of CDI diagnosis, such as body temperature, bowel movements, and bloody stool, were similar in both groups. Intergroup differences in hospitalization within the past 1 year, use of enteral feeding, a history of abdominal surgery, and the types of antimicrobials used were not significant. The FDX group had a significantly higher proportion of patients using potassium-competitive acid blockers (FDX, 36.7% vs. MNZ, 8.0%, *p* < 0.01), whereas the MNZ group had a significantly higher proportion of patients using probiotics before CDI diagnosis (MNZ, 18.7% vs. FDX, 3.3%, *p* = 0.04). The proportions of non-severe cases based on the IDSA/SHEA criteria were 70.7% and 60.0% in the MNZ and FDX groups, respectively (*p* = 0.29). The rates of intensive care unit admission at the time of CDI diagnosis were similar between the two groups.

In the univariate analysis, there was no difference in the primary outcome, i.e., the global cure rate between the MNZ and FDX groups (53.3% vs. 70.0%, *p* = 0.12). Furthermore, the groups did not differ in terms of clinical cure (78.7% (MNZ) vs. 86.7% (FDX), *p* = 0.35), recurrence rate (25.3% (MNZ) vs. 16.7% (FDX), *p* = 0.34), and cause of death within 30 days (1.4% (MNZ) vs. 3.4% (FDX), *p* = 0.50) (Table 2). However, there were significantly more first-line drug changes during CDI treatment in the MNZ group than in the FDX group (18.7% vs. 0.0%, *p* = 0.01). In the MNZ group, 12 patients were switched to VCM or FDX after the initial treatment failed, and 2 were switched to intravenous MNZ. Two patients in the MNZ group were switched to vancomycin due to adverse events. These events included nausea and a decrease in blood count. No adverse events were observed in the FDX group.

The global and non-global cure rates for all patients are compared in Table 3. Although the non-global cure group had a significantly higher proportion of patients with hematological malignancies (*p* = 0.003), there were no significant differences in other factors between the groups. In multivariate analysis of the global cure rate in the total population, FDX treatment, severe CDI, proton pump inhibitor (PPI) use, and age were not associated with a global cure (Table 4).

## 3. Discussion

This retrospective study showed no significant difference between the FDX and oral MNZ treatments in terms of global cure rates in the univariate analysis, although the global cure rate tended to be higher in the FDX treatment group than in the MNZ treatment group. The multivariate analysis did not reveal a significant increase in the global cure rate associated with FDX treatment. Notably, approximately 19% of patients in the MNZ group were switched to other agents during the treatment period because of treatment failure. These findings provide a basis for understanding the comparative efficacy of FDX and oral MNZ in CDI treatment.

The clinical cure and recurrence rates were better in the FDX group than in the MNZ group; however, these differences were not significant. The multivariate analysis revealed that FDX treatment compared to oral MNZ treatment did not significantly affect the global cure rate. Since WBC, which is included in the IDSA/SHEA severity criteria, was significantly higher in the FDX group than in the MNZ group, it is important to consider the possibility that many patients may have been severely ill despite not meeting the severity criteria. Potassium-competitive acid blocker (P-CAB) users were also statistically more prevalent in the FDX group than the MNZ group; P-CABs have been reported to cause more changes in intestinal flora than PPIs [11], which might have influenced the treatment response. Literature searches conducted using PubMed, Google Scholar, and Web of Science did not reveal any direct studies comparing FDX and oral MNZ. However, some studies have compared FDX and VCM, as well as MNZ and VCM, for the treatment of CDI. Several studies have evaluated the clinical efficacy of FDX versus MNZ for CDI treatment through indirect comparisons and meta-network analyses [12,13,14]. A meta-network analysis showed a significant difference in favor of FDX compared with MNZ for sustained cure (clinical cure without recurrence) (odds ratio (OR): 2.39; 95% CI: 1.65–3.47), clinical cure (OR: 1.77; 95% CI: 1.11–2.83), and recurrence (OR: 0.44; 95% CI: 0.27–0.72) [12]. However, another meta-analysis and indirect treatment comparison revealed that FDX led to improved sustained cure rates (clinical cure without recurrence during follow-up; OR: 2.55; 95% CI: 1.44–4.51) and recurrence rates (OR: 0.42; 95% CI: 0.18–0.96) in patients with CDI compared to MNZ. Nevertheless, the intergroup difference in clinical cure was not significant [14]. Additionally, a meta-analysis of real-world data demonstrated no significant differences in recurrence rates between the two groups (OR: 0.71; 95% CI: 0.05–9.47) [13]. While considering differences in study methods and heterogeneity among the articles, it is noteworthy that FDX did not show markedly different results, and there was variability in the findings. Although MNZ treatment has been associated with poorer outcomes compared to VCM treatment, a large cohort study [15] reported that the clinical outcomes achieved with MNZ were comparable to those with VCM if the patients had non-severe CDI and were younger than 65 years, suggesting that oral MNZ may be as effective as other CDI treatment drugs depending on the patient’s background. Therefore, the initial treatment drug for CDI may not necessarily be FDX; however, it may be selected later based on indications, patient background, and economic considerations. 

Our findings suggest the need for caution in the therapeutic management of CDI by using oral MNZ. Although there was no significant difference in the global cure rate between the oral MNZ and FDX groups in this study, it should be considered that the initial treatment failed in approximately 18% of the patients in the MNZ group, resulting in a change in the treatment drug. Previous retrospective cohort studies showed that the rate of switching from MNZ to VCM was 15.9–32% [16,17]. There are several possible reasons for the failure of the initial treatment with oral MNZ. First, the systemic bioavailability of oral MNZ is very high (>90%) [18], with most of it readily absorbed in the gastrointestinal tract, and the drug does not reach particularly high concentrations in the intestinal lumen itself. In contrast, FDX is poorly absorbed from the intestinal tract, and one study reported mean fecal concentrations more than 5000 times the minimum inhibitory concentration of 0.25 μg/mL against *C. difficile* [19]. Notably, the effect of FDX persists on *C. difficile* spores, preventing subsequent growth and toxin production in vitro [20], whereas MNZ does not. Second, the percentage of *C. difficile* strains with reduced susceptibility to MNZ has gradually increased globally [21]. According to a pan-European longitudinal surveillance study, between 2011 and 2012, 18% of *C. difficile* clinical isolates were resistant to MNZ (based on the European Committee on Antimicrobial Susceptibility Testing breakpoint) [22]. However, it is worth noting that *C. difficile* strains resistant to MNZ have never been isolated in Japan [23,24,25], and there have been limited reports of *C. difficile* strains resistant to FDX [26]. Therefore, the effect of MNZ-resistant bacteria on treatment failure in our study was likely small. In cases where the response to initial treatment with oral MNZ is poor, immediate consideration should be given to making necessary changes.

This study has some limitations. First, because this was a single-center retrospective study, the number of patients analyzed was small. Second, it has inherent limitations such as potential selection bias and confounding factors that could not be accounted for. Third, we were unable to assess the differences in the therapeutic efficacy against strains because ribotyping analysis was not performed in this study, although there have been a few reports on ribotype 027 strain isolates in Japan [27]. Despite these limitations, there have been no reports comparing the therapeutic outcomes of FDX and oral MNZ treatments for CDI. Therefore, this study will help clinicians make informed decisions regarding the management of CDI. 

In conclusion, both FDX and oral MNZ demonstrated comparable therapeutic efficacy as initial therapy for CDI. MNZ could potentially serve as a suitable treatment option for initial CDI. However, it necessitates more careful observation since some patients may experience treatment failure and require a change in medication. Further investigations with larger patient cohorts are warranted to thoroughly compare the efficacy of both treatment approaches.

## 4. Materials and Methods

### 4.1. Study Design

This study used a retrospective cohort design and involved the use of the medical records of patients diagnosed with CDI between January 2015 and March 2023 at Aichi Medical University Hospital, a 900-bed tertiary-care hospital located in Aichi, Japan. Ethical approval was granted by the institutional review board of Aichi Medical University (2023-042). Patients ≥2 years old with symptomatic primary CDI who were treated with either FDX or oral MNZ were included. Patients were excluded if they died during CDI treatment or did not show improvement in CDI symptoms and were treated in <10 days or received bezlotoxumab.

The following data were collected from the patient’s electronic medical records: age; sex; body mass index; body temperature and bowel movements at the time of CDI diagnosis; bloody stool; nutrition mode (whether enteral tube feeding was used or not); underlying disease; past hospitalization within 1 year; history of abdominal surgery; intensive care unit admission at the time of CDI diagnosis; laboratory data (white blood cell (WBC) counts and levels of albumin, creatinine (Cr), and C-reactive protein); type of antibiotics administered within 2 months of CDI diagnosis; use of acid suppressants (histamine 2 blockers, PPIs, and potassium-competitive acid blockers), immunosuppressants, anticancer drugs, and prescribed probiotics; severity of CDI; change in anti-CDI medication from the initial treatment; clinical cure; recurrence; and all-cause mortality within 30 days after CDI diagnosis. 

### 4.2. Outcomes

The primary outcome was the difference in the global cure rates of CDI between FDX and MNZ. The secondary outcome was the identification of factors associated with the global cure of patients with CDI as determined using multivariate logistic regression. A systematic review [28] and network meta-analysis [12] showed that advanced age, the use of PPIs, severe CDI, and anti-CDI treatment were associated with poor CDI prognosis. Therefore, these factors were included as explanatory variables. In addition, the rate of medication change due to initial treatment failure and incidence rates of clinical cure, recurrence, adverse effects, and all-cause mortality within 30 days were recorded.

### 4.3. Variable Definition

CDI was defined as follows: (1) the presence of ≥3 diarrheal bowel movements (type 5–7 stool on the Bristol Stool Scale) in the 24 h preceding stool collection, or diarrhea plus patient-reported abdominal pain or cramping, and (2) a positive result for *C. difficile* toxins in a rapid immunoenzyme test for glutamate dehydrogenase (GDH) and a toxin assay (C. DIFF QUIK CHEK COMPLETE; TechLab, Blacksburg, VA, USA or GE Test Immunochromato-CD GDH/TOX; Nissui Pharmaceutical Co., Ltd., Tokyo, Japan) or polymerase chain reaction of the toxin B gene using the Cepheid GeneXpert *C. difficile* Assay (Beckman Coulter Inc., Tokyo, Japan). The C. DIFF QUIK CHEK COMPLETE kit (TechLab) was used between January 2015 and March 2022, whereas the GE test immunochromatography-CD GDH/TOX was used from March 2023. Severe CDI was defined as a WBC count of >15,000 cells/mL or serum Cr level of ≥1.5 mg/dL based on the guidelines released by the IDSA and SHEA [5]. Global cure was defined as clinical cure, no recurrence, and no change in medication owing to a poor response to the initial treatment during the treatment period. We defined the global cure rate as the percentage of patients who completed treatment with MNZ and FDX and met the criteria for global cure. Clinical cure was defined as an improvement in stool characteristics within 2 days of the end of CDI treatment. Recurrent CDI was defined as the re-administration of the initial treatment for diarrhea and a confirmatory positive test up to 8 weeks after the treatment of the initial CDI episode.

### 4.4. Statistical Analysis 

Discrete variables such as age are expressed as medians and interquartile ranges (IQRs). Mann–Whitney *U* and Fisher’s exact tests were used for continuous and two categorical variables, respectively. The significance level was set at 0.05, and a *p*-value < 0.05 was considered statistically significant. Stata version 14.2 (STATA Inc., College Station, TX, USA) was used for statistical analyses.

## Figures and Tables

**Figure 1 antibiotics-12-01323-f001:**
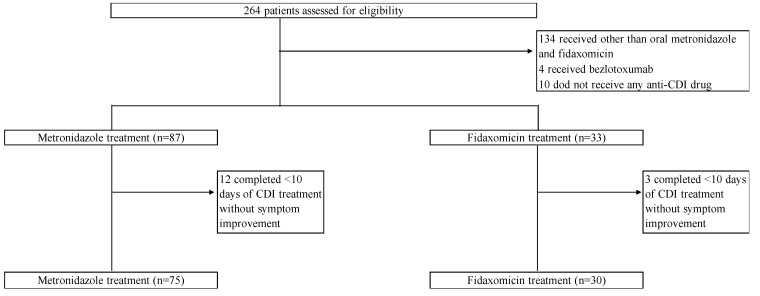
Study flow chart.

**Table 1 antibiotics-12-01323-t001:** Baseline characteristics of patients having CDI with and without recurrence.

Variables	Total (n = 105)	Metronidazole Group (n = 75)	Fidaxomicin Group (n = 30)	*p-*Value
Age (years), median (IQR)	76 (68–83)	76 (70–85)	76 (57–81)	0.17
Female sex, no. (%)	49 (46.7)	36 (48.0)	13 (43.3)	0.83
Body mass index, median (IQR)	18.4 (16.4–21.3)	18.3 (16.4–20.5)	18.5 (16.8–22.0)	0.26
Temperature at the time of CDI diagnosis, median (IQR)	37.6 (37.1–38.2)	37.7 (37.1–38.2)	37.6 (37.1–38.2)	0.82
Bowel movements at the time of CDI diagnosis, median (IQR)	4 (3–7)	4 (3–7)	4 (3–6)	0.93
Bloody stool at the time of CDI diagnosis, median (IQR)	12 (11.4)	10 (13.3)	2 (6.7)	0.33
Comorbidities, no. (%)				
Diabetes mellitus	36 (34.3)	24 (32.0)	12 (40.0)	0.50
Chronic kidney disease	28 (25.7)	16 (21.3)	12 (40.0)	0.09
Heart failure/ischemic heart disease	25 (23.8)	17 (22.7)	8 (26.7)	0.80
Chronic liver disease	2 (1.9)	1 (1.3)	1 (3.3)	0.50
Chronic obstructive pulmonary disease	7 (6.7)	5 (6.7)	2 (6.7)	1.00
Cerebrovascular disease	25 (23.8)	21 (28.0)	4 (13.3)	0.13
Inflammatory bowel disease	4 (3.8)	3 (4.0)	1 (3.3)	1.00
Solid malignancy	26 (24.8)	17 (22.7)	9 (30.0)	0.46
Hematologic malignancy	6 (5.7)	5 (6.7)	1 (3.3)	0.67
Enteral feeding, no. (%)	24 (22.9)	20 (26.7)	4 (13.3)	0.20
Past hospitalization within 1 year, no. (%)	61 (58.1)	40 (53.3)	21 (70.0)	0.13
History of abdominal surgery, no. (%)	23 (21.9)	13 (17.3)	10 (33.3)	0.12
ICU admission at the time of CDI diagnosis, no. (%)	10 (9.5)	8 (19.7)	2 (6.7)	0.72
Non-severe CDI, no. (%)				
IDSA/SHEA criteria	71 (67.6)	53 (70.7)	18 (60.0)	0.35
Laboratory data, median (IQR)				
White blood cell count (/μL)	9100 (5500–12,700)	8000 (5000–11,600)	10,550 (5800–14,500)	0.05
Albumin (mg/dL)	2.4 (2.1–2.9)	2.4 (2.2–2.8)	2.5 (1.6–3.1)	0.98
Creatinine (mg/dL)	0.74 (0.5–1.46)	0.7 (0.48–1)	0.8 (0.54–1.61)	0.30
C-reactive protein (mg/dL)	4.1 (1.3–7.0)	4.01 (1.18–7.05)	4.44 (1.27–6.78)	0.44
Antibiotics, no. (%)				
Penicillin	3 (2.9)	2 (2.7)	1 (3.3)	1.00
Cephalosporin	52 (49.5)	33 (44.0)	19 (63.0)	0.09
Carbapenem	24 (22.9)	17 (22.7)	7 (23.3)	1.00
Fluoroquinolone	13 (12.4)	8 (10.7)	5 (16.7)	0.51
Clindamycin	1 (0.9)	1 (1.3)	0 (0.0)	1.00
β-Lactam/β-Lactamase inhibitor	60 (57.1)	44 (58.7)	16 (53.3)	0.67
Antiviral agents	6 (5.7)	6 (8.0)	0 (0.0)	0.18
Antifungal agents	5 (4.8)	5 (6.7)	0 (0.0)	0.32
Concomitant medication use, no. (%)				
PPIs	49 (46.7)	35 (46.7)	14 (46.7)	1.00
H2RAs	5 (4.8)	5 (6.7)	0 (0.0)	0.32
P-CABs	17 (16.2)	6 (8.0)	11 (36.7)	0.001
Immunosuppression therapy	20 (19.0)	13 (17.3)	7 (23.3)	0.58
Anticancer chemotherapy	7 (6.7)	5 (6.7)	2 (6.7)	1.00
Probiotics used before CDI diagnosis, no. (%)	40 (38.1)	27 (36.0)	13 (43.3)	0.51

CDI: *Clostridioides difficile* infection; IQR: interquartile range; ICU: intensive care unit; PPI: proton pump inhibitor; H2RA: histamine 2 blocker; P-CAB: potassium-competitive acid blocker.

**Table 2 antibiotics-12-01323-t002:** Outcomes of metronidazole and fidaxomicin treatments for CDI.

Variables	Metronidazole Group (n = 75)	Fidaxomicin Group (n = 30)	*p-*Value
Global cure, no. (%)	40 (53.3)	21 (70.0)	0.13
Clinical cure, no. (%)	59 (78.7)	26 (86.7)	0.42
Recurrence, no. (%)	19 (25.3)	5 (16.7)	0.44
Change in the initial CDI treatment, no. (%)	14 (18.7)	0 (0.0)	0.01
Adverse effect, no. (%)	2 (2.7)	0 (0.0)	1.00
All-cause mortality within 30 days, no. (%)	1 (1.4)	1 (3.4)	0.49

CDI: *Clostridioides difficile* infection.

**Table 3 antibiotics-12-01323-t003:** Comparison of global cure and non-global cure in patients with CDI.

Variables	Global Cure Group (n = 61)	Non-Global Cure Group (n = 44)	*p-*Value
Age (years), median (IQR)	77 (70–84)	76 (66–86)	0.93
Male sex, no. (%)	36 (59.0)	20 (45.5)	0.23
Body mass index, median (IQR)	18.3 (16.2–20.4)	18.6 (16.1–21.2)	0.76
Temperature at the time of CDI diagnosis, median (IQR)	37.4 (36.3–38.5)	37.9 (37.2–38.6)	0.09
Bowel movements at the time of CDI diagnosis, median (IQR)	4 (2.5–5.5)	5 (2.5–7.5)	0.08
Bloody stool at the time of CDI diagnosis, no. (%)	8 (13.1)	4 (9.1)	0.76
Comorbidities, no. (%)			
Diabetes mellitus	19 (31.1)	17 (38.6)	0.53
Chronic kidney disease	16 (26.2)	12 (27.3)	1.00
Heart failure/ischemic heart disease	15 (24.6)	10 (22.7)	1.00
Chronic liver disease	2 (3.3)	0 (0.0)	0.50
Chronic obstructive pulmonary disease	3 (4.9)	4 (9.1)	0.45
Cerebrovascular disease	14 (23.0)	11 (25.0)	0.82
Inflammatory bowel disease	3 (4.9)	1 (2.3)	0.64
Solid malignancy	14 (23.0)	12 (27.3)	0.65
Hematologic malignancy	0 (0.0)	6 (13.6)	0.004
Enteral feeding, no. (%)	10 (16.4)	51 (83.6)	0.10
Past hospitalization within 1 year, no. (%)	35 (57.4)	26 (59.1)	1.00
History of abdominal surgery, no. (%)	14 (23.0)	9 (20.5)	0.82
ICU admission at the time of CDI diagnosis, no. (%)	5 (50.0)	56 (91.8)	0.74
Severe CDI, no. (%)	18 (29.5)	16 (36.4)	0.53
Antibiotics, no. (%)			
Penicillin	1 (1.7)	2 (4.5)	0.57
Cephalosporin	31 (59.6)	30 (56.6)	0.84
Carbapenem	10 (41.7)	51 (63.0)	0.10
Fluoroquinolone	7 (11.5)	6 (13.6)	0.77
Clindamycin	0 (0.0)	1 (2.3)	0.42
β-Lactam/β-Lactamase inhibitor	36 (59.0)	24 (54.5)	0.69
Antiviral agents	4 (6.6)	2 (4.5)	1.00
Antifungal agents	2 (3.3)	3 (6.8)	0.65
Concomitant medication use, no. (%)			
PPIs	30 (49.2)	19 (43.2)	0.56
H2RAs	3 (4.9)	2 (4.5)	1.00
P-CAB	12 (19.7)	5 (11.4)	0.29
Immunosuppression therapy	10 (16.4)	10 (22.7)	0.46
Anticancer chemotherapy	3 (4.9)	4 (9.1)	0.45
Probiotics, no. (%)	21 (34.4)	19 (43.2)	0.42
FDX treatment, no. (%)	21 (34.4)	9 (20.5)	0.13

CDI: *Clostridioides difficile* infection; IQR: interquartile range; ICU: intensive care unit; PPI: proton pump inhibitor; H2RA: histamine 2 blocker; P-CAB: potassium-competitive acid blocker.

**Table 4 antibiotics-12-01323-t004:** Multivariable analysis of factors associated with the global cure of *Clostridioides difficile* infection.

Variable	Odds Ratio	95% CI	*p*-Value
Fidaxomicin treatment	1.49	0.94–2.37	0.09
Age	1.01	0.98–1.04	0.66
Severe *Clostridioides difficile* infection	0.68	0.29–1.58	0.37
Use of proton pump inhibitor	1.24	0.56–2.77	0.6

## Data Availability

The data presented in this study are available on request from the corresponding author.

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
