# Peer review of "Clinical Efficacy of Fidaxomicin and Oral Metronidazole for Treating Clostridioides difficile Infection and the Associated Recurrence Rate: A Retrospective Cohort Study"

_antibiotics, 2023, doi:10.3390/antibiotics12081323_

Round 1
Reviewer 1 Report
1. I value the opportunity to review this interesting manuscript. The aim of the paper entitled “Clinical efficacy of fidaxomicin and oral metronidazole for treating Clostridioides difficileinfection and the associated recurrence rate: A retrospective cohort study” may appropriate for consideration of publication in the Antibiotic. However, there are some concern about this manuscript.
2. Despite the limited number of references, the data presented in this manuscript holds scientific significance and is noteworthy. The contemporary approach has led to its citation in 63.64% (14/22) of recent publications over the previous five years, with varying frequency each year from 2018 to 2022 and no citations yet in 2023.However, I suggest considering the inclusion of additional citations to strengthen the manuscript's supporting evidence.
3. Kindly arrange the listed keywords in alphabetical order to ensure a standardized and consistent presentation.
4. The introduction of the manuscript could be strengthened by further highlighting the importance of using metronidazole, and I would recommend the author to consider this
5. I kindly suggest that the inclusion criteria for Clostridioides difficileinfection be further specified in the manuscript. Providing additional details regarding the inclusion criteria would likely improve the study's clarity and impact in the field.
6. To ensure the validity and reliability of statistical analysis, I suggest conducting a normality check of the data before performing parametric or nonparametric tests.
7. For categorical data with sparse cell counts, choosing the appropriate statistical analysis for two groups requires careful consideration. Fisher's exact test may be more appropriate than the chi-square test as it is an exact hypothesis testing method that has no underlying assumptions concerning sample size. It is particularly useful when working with low-frequency cell counts.
8. For enhanced clarity and readability, it is suggested to include three horizontal lines in table 4 to distinctly separate the table's header, body, and footer sections.
9. I recommend discussing variables that demonstrate statistically significant differences, such as comorbidities with chronic kidney disease, white blood cell count, and concomitant medication use of P-CABs. Including a discussion of these variables would likely provide valuable insights that could enhance the manuscript's overall impact.
10. I respectfully request that the authors provide further details regarding the pharmacokinetic and pharmacodynamic effects of both drugs, if possible. As a deeper exploration of these effects may offer valuable insights into the identified disparities in epidemiological outcomes, I believe it would be beneficial to include such information in the manuscript.
11. To improve the clarity and effectiveness of this manuscript, it is recommended to provide a more specific conclusion section. It would be beneficial to explicitly state the primary revelations derived from the significant content gathered and offer practical suggestions for their clinical applications. Furthermore, including a recommendation for further research at the end of the conclusion section would effectively encourage future investigations in this area.
12. The privilege to have reviewed this manuscript is deeply appreciated. While acknowledging the significance and potential contributions of the topic presented, it is my expert opinion that more clarity and impetus is needed to advance the work. I remain hopeful that the provided recommendations will be beneficial to the authors in enhancing their manuscript in preparation for publication.
Author Response
■Despite the limited number of references, the data presented in this manuscript holds scientific significance and is noteworthy. The contemporary approach has led to its citation in 63.64% (14/22) of recent publications over the previous five years, with varying frequency each year from 2018 to 2022 and no citations yet in 2023.However, I suggest considering the inclusion of additional citations to strengthen the manuscript's supporting evidence.
Response: Thank you for your valuable comment. As per your suggestion, we have cited an epidemiology-focused article published in 2023 (Reference 1).
■Kindly arrange the listed keywords in alphabetical order to ensure a standardized and consistent presentation.
Response: Thank you for your valuable comment. We have arranged the listed keywords in alphabetical order.
■The introduction of the manuscript could be strengthened by further highlighting the importance of using metronidazole, and I would recommend the author to consider this.
Response: Thank you for your valuable comment. We have emphasized the importance of using metronidazole.
■ I kindly suggest that the inclusion criteria for Clostridioides difficile infection be further specified in the manuscript. Providing additional details regarding the inclusion criteria would likely improve the study's clarity and impact in the field.
Response: Thank you for your valuable comment. We have included a more specific definition of CDI.
■ To ensure the validity and reliability of statistical analysis, I suggest conducting a normality check of the data before performing parametric or nonparametric tests.
Response: Thank you for your valuable suggestion. In this study, the nonparametric method, Mann–Whitney's U test, was employed to compare the two groups. This method was chosen because it does not depend on the distribution or type of data, making it suitable for parametric data as well without compromising its reliability or validity.
■For categorical data with sparse cell counts, choosing the appropriate statistical analysis for two groups requires careful consideration. Fisher's exact test may be more appropriate than the chi-square test as it is an exact hypothesis testing method that has no underlying assumptions concerning sample size. It is particularly useful when working with low-frequency cell counts.
Response: Thank you for your valuable comment. As per your suggestion, we have utilized Fisher’s exact test to re-evaluate the categorical data.
■ For enhanced clarity and readability, it is suggested to include three horizontal lines in table 4 to distinctly separate the table's header, body, and footer sections.
Response: We have implemented the required modifications as instructed.
■ I recommend discussing variables that demonstrate statistically significant differences, such as comorbidities with chronic kidney disease, white blood cell count, and concomitant medication use of P-CABs. Including a discussion of these variables would likely provide valuable insights that could enhance the manuscript's overall impact.
Response: Thank you for your valuable comment. In the Discussion section, we examined the WBC and P-CABs, which showed significant differences between the two groups. However, no additional information on CKD was included because Fisher's exact test did not reveal a significant difference.
■I respectfully request that the authors provide further details regarding the pharmacokinetic and pharmacodynamic effects of both drugs, if possible. As a deeper exploration of these effects may offer valuable insights into the identified disparities in epidemiological outcomes, I believe it would be beneficial to include such information in the manuscript.
Response: Thank you for your valuable comment. In the Discussion section, we presented information regarding the pharmacokinetic and pharmacodynamic effects of MNZ and FDX.
■ To improve the clarity and effectiveness of this manuscript, it is recommended to provide a more specific conclusion section. It would be beneficial to explicitly state the primary revelations derived from the significant content gathered and offer practical suggestions for their clinical applications. Furthermore, including a recommendation for further research at the end of the conclusion section would effectively encourage future investigations in this area.
Response: Thank you for your valuable comment. We have revised our conclusions as follows:
“In conclusion, both FDX and oral MNZ demonstrated comparable therapeutic efficacy as initial therapy for CDI. MNZ could potentially serve as a suitable treatment option for initial CDI. However, it necessitates more careful observation since some patients may experience treatment failure and require a change in medication. Further investigations with larger patient cohorts are warranted to thoroughly compare the efficacy of both treatment approaches.”
Reviewer 2 Report
1. Authors may elaborate the clinical manifestation of Chloridoids difficile infection (CDI).
2. The authors should also collect the patient data who had shown adverse reaction to both MNZ and FDX.
3. Why was chi-square test used for statistical analysis?
4. Evaluation of safety of the use of MNZ and FDX should be shown.
5. The authors must show if there is any difference in results between male and female patients of both the groups.
6. The authors should describe the global cure rate of CDI being considered.
Author Response
Responses to Reviewer 2 Comments
■Authors may elaborate the clinical manifestation of Chloridoids difficile infection (CDI).
Response: Thank you for your valuable comment. As per your suggestion, we have included a more specific definition of CDI.
■The authors should also collect the patient data who had shown adverse reaction to both MNZ and FDX.
Response: We have evaluated and demonstrated the adverse effects of both MNZ and FDX.
■Why was chi-square test used for statistical analysis?
Response: Thank you for your insightful question. We have reanalyzed the data using Fisher's exact test.
■Evaluation of safety of the use of MNZ and FDX should be shown.
Response: We have evaluated and demonstrated the adverse effects of both MNZ and FDX.
■The authors must show if there is any difference in results between male and female patients of both the groups.
Response: Thank you for your valuable comment. We have examined global cure rates by gender but found no significant results.
■The authors should describe the global cure rate of CDI being considered.
Response: We defined the global cure as achieving clinical cure, experiencing no recurrence, and not requiring a change in medication due to a poor response to the initial treatment during the treatment period. The global cure rate was calculated as the percentage of patients who completed treatment with MNZ and FDX and met the criteria for global cure.
Reviewer 3 Report
The study titled "Comparing the clinical efficacy of fidaxomicin and oral metronidazole for treating Clostridioides difficile infection and associated recurrence rates: A retrospective cohort study" aimed to assess and compare the effectiveness of fidaxomicin (FDX) and oral metronidazole (MNZ) in treating Clostridioides difficile infection (CDI) and their respective recurrence rates. The authors provided a clear rationale for this comparison, particularly in the context where fidaxomicin was approved for CDI treatment.
However, several important comments were raised regarding the study:
Comments:
1. The authors mentioned that "....75 and 30 were assigned to the oral MNZ and FDX groups, respectively." It is necessary to clarify the rationale or method used for assigning patients to receive oral MNZ or FDX in each group.
2. The study did not explicitly report how CDI was defined and diagnosed, potentially affecting the accuracy and consistency of the inclusion and exclusion criteria.
3. A power calculation to determine the required sample size for detecting a statistically significant difference between FDX and MNZ treatments was not conducted, which could impact the study's ability to draw robust conclusions.
4. The quality of data sources and the validity of outcome measures were not assessed, which might introduce measurement errors and biases to the study.
5. The study did not conduct sensitivity or subgroup analyses to test the robustness and generalizability of the results, which could provide valuable insights into the reliability of the findings.
6. No comparison was made between the study's results and those of other relevant studies, and a systematic review of the literature on FDX and MNZ for CDI treatment was not provided.
7. The authors did not discuss their findings' ethical, practical, or theoretical implications for clinical practice and policy-making.
8. Besides figure 1, there are no other figures in the manuscript. Some information from the tables could be presented in figures, enhancing reader understanding.
9. Ribotyping analysis, which could have provided additional insights into the strains causing CDI, was not performed in the study.
Author Response
Responses to Reviewer 3 Comments
■The authors mentioned that "....75 and 30 were assigned to the oral MNZ and FDX groups, respectively." It is necessary to clarify the rationale or method used for assigning patients to receive oral MNZ or FDX in each group.
Response: We have modified Figure 1 to reflect the required edit.
■The study did not explicitly report how CDI was defined and diagnosed, potentially affecting the accuracy and consistency of the inclusion and exclusion criteria.
Response: We have included a more specific definition of CDI.
■ A power calculation to determine the required sample size for detecting a statistically significant difference between FDX and MNZ treatments was not conducted, which could impact the study's ability to draw robust conclusions.
Response: Thank you for your valuable suggestion. As this study is retrospective and observational in nature, it is not suitable to conduct sample size calculations. By power calculation, do you mean to calculate the statistical power for each of the outcomes presented in Table 2, for example?
■ The quality of data sources and the validity of outcome measures were not assessed, which might introduce measurement errors and biases to the study.
Response: Thank you for your valuable suggestion. The possibility of selection bias has been reflected in the limitations portion in the manuscript.
■The study did not conduct sensitivity or subgroup analyses to test the robustness and generalizability of the results, which could provide valuable insights into the reliability of the findings.
Response: Thank you for your valuable suggestion. Stratified analysis by subgroup was not conducted. Instead of assessing the robustness and generalizability of the results, we acknowledged the potential for distorting the results through subsequent analyses.
- No comparison was made between the study's results and those of other relevant studies, and a systematic review of the literature on FDX and MNZ for CDI treatment was not provided.
Response: Literature searches using PubMed, Google Scholar, and Web of Science did not identify any direct studies comparing FDX and oral MNZ. Hence, we did not provide a systematic review of the literature on FDX and MNZ for CDI treatment.
■ The authors did not discuss their findings' ethical, practical, or theoretical implications for clinical practice and policy-making.
Response: Thank you for your valuable comment. To address your concern, we have revised our conclusions as follows:
“In conclusion, both FDX and oral MNZ demonstrated comparable therapeutic efficacy as initial therapy for CDI. MNZ could potentially serve as a suitable treatment option for initial CDI. However, it necessitates more careful observation since some patients may experience treatment failure and require a change in medication. Further investigations with larger patient cohorts are warranted to thoroughly compare the efficacy of both treatment approaches.”
■ Besides Figure 1, there are no other figures in the manuscript. Some information from the tables could be presented in figures, enhancing reader understanding.
Response: Thank you for your valuable suggestion. The results of the table are already summarized in the text, and I prefer to present the results as they are without adding new figures to avoid duplicating information.
■ Ribotyping analysis, which could have provided additional insights into the strains causing CDI, was not performed in the study.
Response: Thank you for your comments. As you pointed out, due to the retrospective nature of the study, we were unable to conduct ribotyping analysis. This has been reflected in the Discussion section in the manuscript.
Round 2
Reviewer 1 Report
The authors have satisfactorily addressed most of my concerns. This revision has also significantly improved the manuscript.
Reviewer 2 Report
The revised manuscript may be accepted for publication.
Reviewer 3 Report
The authors have addressed my concerns and suggestions thoughtfully and comprehensively.
They have provided detailed explanations for the changes they made in response to my feedback, and their revisions have significantly improved the quality and clarity of the manuscript.